# Histological Variants of Squamous and Basal Cell Carcinoma in Squamates and Chelonians: A Comprehensive Classification

**DOI:** 10.3390/ani13081327

**Published:** 2023-04-12

**Authors:** Ferran Solanes Vilanova, Tom Hellebuyck, Koen Chiers

**Affiliations:** Department of Pathobiology, Pharmacology and Zoological Medicine, Faculty of Veterinary Medicine, Ghent University, Salisburylaan 133, B-9820 Merelbeke, Belgium

**Keywords:** basal cell carcinoma, immunohistochemistry, neoplasm, reptiles, squamous cell carcinoma

## Abstract

**Simple Summary:**

The present study investigated the histological characteristics of 35 tumors from 21 lizards, 1 snake, 10 tortoises and 3 turtles that were initially diagnosed as squamous or basal cell carcinoma. Based on in-depth re-evaluation of the tissue characteristics, eight tumors initially diagnosed as squamous cell carcinoma were re-classified as basal cell carcinomas and three squamous cell carcinomas proved to be non-neoplastic lesions. All squamous and basal cell carcinomas were classified into distinct histological variants. To date, basal cell carcinomas have only been described in two reptile species. In the present study, basal cell carcinomas were diagnosed in seven additional species. While immunohistochemical staining with cyclooxygenase-2 and E-cadherin showed significant differences between the examined squamous and basal cell carcinomas, no immunoreactivity was observed for epithelial antigen clone Ber-EP4 and epithelial membrane antigen. The results of this study provide a proposal classification that allows the differentiation of squamous and basal cell carcinoma and their histological variants in squamates and chelonians.

**Abstract:**

In the present study, the histological characteristics of squamous cell carcinomas (SCCs) and basal cell carcinomas (BCCs) obtained from 22 squamate and 13 chelonian species were retrospectively evaluated. While the examined tissues were originally diagnosed as 28 SCCs and 7 BCCs based on histological evaluation by a specialty diagnostic service, eight SCCs could be re-classified as BCCs and three SCCs proved to be non-neoplastic lesions. In addition, all SCCs and BCCs were classified into distinct histological variants. The SCCs could be categorized as one SCC in situ, three moderately differentiated SCCs, seven well-differentiated SCCs, and six keratoacanthomas. BCCs were classified as five solid BCCs, four infiltrating BCCs, five keratotic BCCs, and one basosquamous cell carcinoma. In addition, the present study reports the occurrence of BCCs in seven reptile species for the first time. In contrast to what has been documented in humans, IHC staining with the commercially available epithelial membrane antigen and epithelial antigen clone Ber-EP4 does not allow differentiation of SCCs from BCCs in reptiles, while cyclooxygenase-2 and E-cadherin staining seem to have discriminating potential. Although the gross pathological features of the examined SCCs and BCCs were highly similar, each tumor could be unequivocally assigned to a distinct histological variant according to the observed histological characteristics. Based on the results of this study, a histopathological classification for SCCs and BCCs is proposed, allowing accurate identification and differentiation of SCCs and BCCs and their histological variants in the examined reptile species. Presumably, BCCs are severely underdiagnosed in squamates and chelonians.

## 1. Introduction

Neoplasms are frequently encountered in the practice of reptile medicine, although they were once considered uncommon [1]. Most data about the occurrence of neoplasms in captive reptiles originate from specialty diagnostic services [1,2,3,4,5], and considerable variation in prevalence data, ranging from 9.8% to 26%, is reported [1,2,3,4,5,6,7]. In general, neoplasms are more frequently observed in snakes and lizards in comparison to chelonians and crocodilians [1,2,4,5,6], with the integumentary, hepatic, and musculoskeletal systems being the most commonly affected sites [1,2,4,5,6]. Reports of skin tumors in reptiles are largely derived from single cases, and mainly comprise squamous cell carcinomas (SCCs), papillomas, and chromatophoromas [1,2,3,5]. The increasing number of neoplastic disorders that are being diagnosed in captive reptiles can at least be partly attributed to the fact that reptile owners more readily seek veterinary advice as well as to the increasing availability and use of appropriate diagnostic tools. In addition, the increasing lifespan of reptile pets as well as predisposing environmental and genetic factors may also contribute to the seemingly increasing prevalence of neoplastic disorders [3,5,6,8,9,10]. Nevertheless, the diagnosis of neoplasms in reptile patients can easily be missed at initial presentation as associated clinical signs are often non-specific [4,5].

Basal cell carcinomas (BCCs) and SCCs account for 96% of skin neoplasms in humans. Human BCCs are the most commonly diagnosed non-melanoma skin tumors and are diagnosed three to four times more often than SCCs [11]. While SCCs are the most common skins neoplasm in cats and the second most common in dogs [12], most BCCs in cats and dogs have been re-classified as benign trichoblastomas and apocrine ductular adenomas, respectively [12,13]. Consequently, true BCCs are considered to be relatively rare neoplasms in these conventional domestic animals [12,13]. Furthermore, SCCs are among the most common integumentary neoplasms in reptilian species, while BCCs are rarely documented in these taxa. At present, reports of BCCs in reptile species are limited to a savannah monitor (*Varanus exanthematicus*), and two Hermann’s tortoises (*Testudo hermanni*) [5,14]. Although the skin and oral cavity seem to be predilection sites for both SCCs and BCCs in reptiles, SCCs originating near the mucocutaneous junction (MCJ) are also frequently reported, particularly in bearded dragons (*Pogona vitticeps*) [3]. Recently, keratoacanthoma (KA) has been described as a new histological variant of dermal SCCs in lizards with a presumed species predisposition in panther chameleons (*Furcifer pardalis*) that often shows a multicentric distribution, including involvement of the MCJ of the eyelid [8].

The discrimination of dermal SCCs and BCCs and their histological variants is highly important towards prognosis estimation and establishing appropriate treatment protocols as the associated invasiveness, recurrence rate, and metastatic potential are strongly correlated with the involved histological variant [15,16]. Histological variants of SCCs and BCCs have been characterized in humans, dogs and cats and classifications have been established [12,15,17]. Several classifications for human SCCs and BCCs were adopted by the International Agency for Research on Cancer (IARC) and included in the World Health Organization (WHO) classification for skin tumors [18]. However, the differentiation and histological classification of certain variants remain challenging and, in some cases, even controversial [12,18]. Nevertheless, correct identification and classification of SCCs and BCC variants are essential towards the establishment of the most effective treatment protocols [12,19,20].

At present, commonly observed high-risk histological variants of SCCs include acantholytic SCCs, desmoplastic SCCs, and spindle cell SCCs, and low-risk histological variants include verrucous SCC and KA. For human BCCs, infiltrating BCCs and BSCCs represent high-risk histological variants, and solid BCCs, keratotic BCCs, and superficial BCCs represent low-risk histological variants [18]. While SCCs with varying degrees of differentiation are commonly observed in reptiles as well as birds [21,22,23,24], studies focused on in-depth comparative assessment of the histological characteristics of these neoplasms that allow the establishment of a classification system, are lacking.

In human medicine, immunohistochemistry (IHC) has proven to be an essential tool to differentiate certain histological variants of SCCs from BCCs [25,26]. In addition, IHC aids in tumor staging, selecting optimal treatment protocols, and identifying genetic variants in humans, dogs and cats [25,26,27,28,29,30,31]. Although two mammalian alpha-keratin markers were successfully used to characterize SCCs in loggerhead sea turtles (*Caretta caretta*), the limited availability of reptilian monoclonal antibodies and the lack of commercially available antibodies that cross-react with reptilian tissue continue to hamper the use of IHC in reptiles [7,32,33,34].

Taking into account the challenges that have been and are still encountered in humans and conventional pets towards the correct discrimination, identification, and classification of histological variants of SCCs and BCCs [35], the present study aims to provide a basis for the correct histological characterization and classification of SCCs and BCCs and their histological variants in squamates and chelonians.

## 2. Materials and Methods

### 2.1. Tissues

Formalin-fixed and paraffin-embedded tissues from lesions obtained in 35 unrelated, captive reptiles (22 squamates and 13 chelonians) that were presented at a veterinary teaching hospital between 2010 and 2022 were included. The paraffin-embedded tissues were selected based on their initial diagnosis as SCCs or BCCs by a specialty diagnostic service following routine histological examination (Table 1). The tumors either originated from the skin, the epidermis of the shell, the MCJ of the eyelid, or the oral mucosa. All tissues were collected antemortem following excisional biopsy or in toto during surgical resection of the lesion.

### 2.2. Histopathology

Paraffin-embedded blocks were cut into 5-µm thick sections and stained with haematoxylin and eosin (HE). All sections were re-evaluated and neoplasms were further characterized. Mitotic figures were counted in 10 high-power fields (HPF) in randomly chosen areas and mean numbers were calculated. The mitotic index was graded as low (fewer or 2 mitoses per 10 HPFs), moderate (3 to 4 mitoses per 10 HPFs), or high (5 or more mitoses per 10 HPFs). The degree of nuclear atypia was graded as mild, moderate, or marked if less than 30%, between 30% and 60%, or more than 60% of the neoplastic cells had nuclear atypia, respectively.

### 2.3. Immunohistochemistry

IHC staining for epithelial antigen clone Ber-EP4, epithelial membrane antigen (EMA) clone E29, cyclooxygenase-2 (COX-2) clone 33, E-cadherin clone NCH-38, and cytokeratin (Pan-CK) clones AE1/AE3 were performed in all tumors. Tissues were cut into 5-μm thick sections and prepared on 3-aminopropyltriethoxysilane-coated (APES) slides. The slides were then deparaffinised and rehydrated in xylene and decreasing concentrations of alcohol in H2O (100, 96, 50, and 100% H_2_O, respectively).

Antigen retrieval was performed by immersion in citrate-buffered (0.01 M, pH 6) distilled water and microwave treatment for 3.5 min at 850 W and 10 min at 450 W. Next, slides were allowed to cool down for 20 min and incubated with H2O2 (S202386-2, Agilent, Santa Clara, CA, USA) at room temperature for 5 min. Subsequently, slides were incubated with mouse primary monoclonal COX-2 (1/20, 610204, BD Biosciences, San José, CA, USA)/E-cadherin (1/100, M3612, Agilent, Santa Clara, CA, USA)/Pan-CK (1/50, M3515, Agilent, Santa Clara, CA, USA)/epithelial antigen clone Ber-EP4 (1/10, M0804, Agilent, Santa Clara, CA, USA)/EMA (1/10, M0613, Agilent) antibody at room temperature for 30 min with an antibody diluent solution with background-reducing components (S302283-2, Agilent). After incubation with a polymer-based secondary anti-mouse antibody (K400111, Agilent, Santa Clara, CA, USA) at room temperature for 30 min, visualization was performed in a 3.3-diaminobenzidine solution (K346811, Agilent) at room temperature for 5 min. Cell nuclei were counterstained with haematoxylin, rinsed in tap water, dehydrated and coverslips applied. Between all steps, the sections were washed extensively and repeatedly with phosphate-buffered saline.

Negative controls consisted of omitting the primary antibody in normal skin samples from a dog and a bearded dragon. Normal skin samples from a dog were used as positive controls for E-cadherin, COX-2 and pan-CK and mammary gland tissues were used as positive controls for epithelial antigen clone Ber-EP4 and EMA.

To evaluate the expression of epithelial antigen clone Ber-EP4, EMA, COX-2, E-cadherin, and Pan-CK, an immunoreactive score system (IRS), based on the percentage of positive cells and intensity of staining according to Fedchenko et al. [36], was used (Table 2).

### 2.4. Statistical Analysis

Statistical analysis was performed by using the SPSS statistical software (IBM SPSS Statistics version 27.0). A *p*-value < 0.05 with a 95% confidence interval was considered statistically significant.

The Mann–Whitney U test was used to compare the median ranks of continuous variables between two independent groups. The Kruskal–Wallis test was used to compare the median ranks of continuous variables between three and more independent groups, and the Friedman test was used to compare the median ranks of continuous variables between three and more related groups.

## 3. Results

### 3.1. Histological Re-Classification

Based on the retrospective histological characterization of tissues obtained from 35 individual reptile patients, 3 tissues that were initially diagnosed as SCCs proved to be non-neoplastic lesions and 8 SCCs were re-classified as BCCs. The three non-neoplastic lesions consisted of a cystic mass lined by pseudostratified ciliated epithelium, pyogranulomatous dermatitis with irregular epidermal hyperplasia, and gingival fibrous hyperplasia. In total, 17 out of the remaining 32 tissues were identified as SCCs (53.1%) and 15 were identified as BCCs (46.9%) (Table 1). A total of 13 out of 17 (76.5%) SCCs were obtained from squamates and 4 out of 17 (23.5%) from chelonians, while 7 out of 15 (46.7%) BCCs were obtained from squamates and 8 out of 15 (53.3%) from chelonians.

Sixteen of the SCCs (94.1%) had a dermal origin and one (5.9%) originated from the oral mucosa. Eleven dermal SCCs originated from the skin, one from the epidermis of the shell and four from the MCJ of the eyelid (Table 1). A total of 12 out of 15 BCCs (80%) had a dermal origin and 3 (20%) originated from the oral mucosa. Of the 12 dermal BCCs, 7 originated from the skin, 4 from the epidermis of the shell, and 1 from the MCJ of the eyelid.

### 3.2. Squamous Cell Carcinoma and Its Histological Variants

Generally, all SCCs could be defined as malignant neoplasms originating from the stratified squamous epithelium of the oral mucosa, MCJ of the eyelid or epidermis that presented as irregular proliferations of tumor cells with various degrees of differentiation. The neoplastic cells were often large with abundant eosinophilic cytoplasm and large nuclei, resulting in a low nuclear-to-cytoplasm ratio. In a few SCCs, areas of necrosis and cystic degeneration were noted. A variable degree of synchronous differentiation of peripheral basal-type cells to central squamous epithelial cells was observed. The most advanced stage of differentiation resulted in advanced keratinization, presenting as keratin pearls. Consequently, the presence and number of keratin pearls could be related to the degree of differentiation of the involved neoplasm. In several cases, ulceration of the neoplastic nests with infiltration of inflammatory cells from the dermis into the epidermis or subcutis was present.

The histological SCC variants included 1 SCC in situ (5.9%), 10 conventional SCCs (58.8%) and 6 KA (35.3%). Conventional SCCs could be further classified into two histological grades, representing either well-differentiated SCCs (7 cases; 41.2%) or moderately differentiated SCCs (3 cases; 17.6%). An overview of the histological variants with their mitotic index and degree of nuclear atypia are provided in Table 3.

The SCC in situ was characterized by epidermal dysplasia with enlarged and pleomorphic squamous cells that showed high mitotic activity and mild nuclear atypia, replacing the entire thickness of the epidermis without invading the basal membrane (Figure 1).

Conventional SCCs infiltrated the papillary dermis and, in some cases, the reticular dermis and the subcutis. Seven conventional SCCs were graded as well-differentiated SCCs and were characterized by tumor cells containing slightly enlarged and hyperchromatic nuclei with abundant eosinophilic cytoplasm (Figure 2). They exhibited low mitotic activity, mild to moderate nuclear atypia, and evidence of synchronous differentiation of peripheral basal-type cells to central squamous epithelial cells, which resulted in the formation of extracellular keratin pearls in most cases.

Three conventional SCCs were graded as moderately differentiated SCCs based on a low mitotic index and marked nuclear atypia (Figure 3). Keratin pearl formation in moderately differentiated SCCs was a less prominent feature in comparison to well-differentiated SCCs.

Six SCCs could be unambiguously classified as KAs and were exclusively diagnosed in lizards. They presented as an exo–endophytic, cyst-like epidermal proliferation that creates a crateriform lesion with a central keratinous plug (Figure 4). Areas of pseudoepitheliomatous hyperplasia with minimally infiltrating well-differentiated squamous cells formed folds inside the crater and the adjacent dermis. Mild nuclear atypia and a low mitotic index were invariably present.

### 3.3. Basal Cell Carcinoma and Its Histological Variants

BCCs were identified as malignant proliferations of the skin, the MCJ of the eyelid or the oral mucosa originating from the epidermal basal cells. All BCC histological variants typically contained islands or nests of cuboidal basaloid cells with a central, haphazard, hyperchromatic nuclear arrangement and scant amount of slightly basophilic cytoplasm, resulting in a high nuclear-to-cytoplasm ratio. Keratinization varied according to the histological variant. Peripheral palisade formation and stromal cleft formation was considered a rare and inconsistent histological feature. Sporadic areas of necrosis and cystic degeneration were noted. In several cases, ulceration with infiltration of inflammatory cells from the dermis into the epidermis or subcutis was present. Four distinct histological variants could be identified: solid BCCs (5 cases; 33.3%), keratotic BCCs (5 cases; 33.3%), infiltrating BCCs (4 cases; 26.7%), and basosquamous cell carcinoma (1 case; 6.7%). An overview of the histological variants and their mitotic index and degree of nuclear atypia are provided in Table 3.

Solid BCCs (also referred to as nodular BCCs) were characterized by cords of small polyhedral basaloid cells extending into the dermis that contained small foci of squamous metaplasia (Figure 5). Advanced keratinization with frank extracellular keratin production resulting in the formation of keratin pearls in the squamoid foci was occasionally seen. The mitotic activity, the number of atypical mitotic figures and the degree of nuclear atypia were low, except for one solid BCC that showed high nuclear atypia and moderate mitotic activity.

Keratotic BCCs presented a similar architecture as solid BCCs, but most epithelial islands contained central or peripheral foci of abrupt squamous differentiation with central mature keratinization (Figure 6). In addition, the squamous cells possessed large, vesicular nuclei with small nucleoli and atypical mitotic figures. The neoplastic cells exhibited variable mitotic activity, ranging from low to high, with minimal nuclear atypia.

Infiltrating BCCs were mainly characterized by irregular, narrow (<8 cells thick), and elongated cords of small, atypical basophilic basal tumor cells, without differentiation into squamous epithelium that were deeply infiltrating into the dermis and subcutis (Figure 7). The neoplastic cells showed moderate to marked nuclear atypia, and mitotic activity was consistently observed. Extensive fibroblast proliferation of the dermis was often observed in response to the infiltrating neoplastic cords.

Basosquamous cell carcinomas (BSCCs) (also referred to as metatypical BCCs) presented areas with BCC as well SCC features (Figure 8). Atypical squamous cells formed scattered islands, trabeculae, and nests that often had angular, irregular profiles. Similar to what is seen in conventional SCCs and solid BCCs, a scarce stroma with moderate cellularity surrounded the epithelial structures. Architecturally, the histological characteristics of the BSCCs resembled those of keratotic BCCs but within the squamous component, malignant histological features as observed for SCCs were prominent. Frequently, keratin pearls were present.

### 3.4. Immunohistochemistry

While 70.6% of the SCCs exhibited strong expression levels for E-cadherin, the remaining 29.4% showed moderate expression levels (Table 4). In contrast, 80% of the BCCs showed moderate E-cadherin expression and the remaining 20% showed poor expression levels, without overrepresentation of a particular histological variant. Statistical analysis demonstrated that the observed difference in expression of E-cadherin between SCCs and BCCs was significant (*p* < 0.05), with SCCs generally displaying higher expression levels compared to BCCs (Figure 9). While E-cadherin expression levels were significantly higher in the KAs compared to the well-differentiated SCCs (*p* < 0.05), no significant differences in E-cadherin expression could be observed between the BCC histological variants.

The proportion of SCCs that showed moderate or strong COX-2 expression was higher (58.8% and 41.2%, respectively) than what was observed for the BCCs that almost in all cases showed poor to negative expression (26.7% and 46.7%, respectively). Only in keratotic BCCs, moderate COX-2 expression could be noted (Table 4). Based on the performed statistical analysis, a significantly higher COX-2 expression was observed in the SCCs than in the BCCs (*p* < 0.05) (Figure 10). While KA histological variants exhibited lower COX-2 expression compared to the other SCC variants (*p* < 0.05), the keratotic BCC histological variant exhibited higher expression levels than the other BCC variants (*p* < 0.05).

All SCCs and BCCs described in this study showed strong pan-CK reactivity, which allowed sharp delineation of the neoplastic processes from the surrounding tissue and assessment of the integrity of the basement membrane.

Immunohistochemical analysis utilizing antisera against epithelial antigen clone Ber-EP4 and EMA did not demonstrate positive staining in either the tumor cells or the control skin tissues from reptiles. Nonetheless, positive controls consisting of mammary gland tissue from dogs showed positive immunoperoxidase labelling for epithelial antigen clone Ber-EP4 and EMA.

## 4. Discussion

Taking into account the existing human, canine, and feline classifications, we propose a histological classification for SCCs and BCCs in squamates and chelonians based on the results of the present study. As for humans and conventional pets, such a classification may facilitate adapting the clinical and therapeutic approach according to the involved SCC or BCC histological variant. Considering the existing controversy regarding the definition and classification of certain SCC and BCC histological variants in humans as well as dogs and cats [12,20,21,37,38,39], the proposed classification for squamates and chelonians should be considered as a dynamic concept that needs to be subjected to regular evaluation and revision based on scientific progress and new insights.

Even in humans, accurate identification of BCC and SCC histological variants [8,40], might be challenging because of their similar basic histological in addition to their often highly comparable clinical appearance as well as their mutual predilection sites. Moreover, certain species predispositions are reported in reptiles, notably for certain SCC histological variants in bearded dragons and panther chameleons [3,8,10]. The latter may strongly bias the differential diagnostic approach and the differentiation of SCCs from BCCs as illustrated by the initial misdiagnosis of eight SCCs that were re-classified as five keratotic BCCs, two infiltrative BCCs, and one solid BCC in the present study. As normal reptile skin typically contains higher amounts of keratin due to the presence of alpha and beta keratin epidermal layers in comparison to humans and other mammals [41], especially keratotic BCCs can be easily misdiagnosed as conventional SCCs or BSCCs because of the abundant presence of keratin pearls [12]. For this reason, it was fundamental to classify keratotic BCCs as a distinct histological variant from solid BCCs in the present study. Misdiagnosis of infiltrative BCCs is presumably related to the limited amount of neoplastic basal cells in the typically narrow neoplastic cords of this histological variant [1,2,3]. The solid BCC that was misdiagnosed as SCC originated from the epidermis of the shell. The high concentration of rigid beta-keratin layers in the shell of most chelonian species makes its epidermis particularly hard to process for histological sectioning [42]. This can potentially lead to loss of keratin layers or the architectural pattern of the sample, which are essential for histological characterization of BCCs and SCCs. For the solid BCC in the present case, repeated sectioning of the paraffin-embedded tissue was necessary to obtain a correct histological diagnosis. In conclusion, BCCs are presumably highly underdiagnosed in reptiles because of their misdiagnosis as SCCs and the results of the present study may raise awareness to correctly identify BCCs in reptiles.

Based on the histological examination of SCCs and BCCs from squamates and chelonians, the characteristics observed for all histological variants that were identified in the present study, fully comply with the definitions applied for these histological variants in humans [20]. In addition, KA characteristics fully complied with those described by Solanes et al. [8]. Other commonly observed SCC histological variants in humans, such as acantholytic SCCs, desmoplastic SCCs, spindle cell SCCs, and verrucous SCCs, could not be demonstrated in the present study. Nevertheless, it cannot be excluded that these histological variants may occur in reptiles.

Although poorly differentiated SCCs were not demonstrated in the present study, this third histological grade of conventional SCCs may be encountered in reptiles. Histological characteristics of poorly differentiated SCCs in humans include enlarged, pleomorphic nuclei with asynchronous differentiation from basal to central squamous cells, marked nuclear atypia, and frequent mitoses [20,22]. Keratin pearls are a highly unusual finding in poorly differentiated SCCs.

At present, infiltrating BCCs and sclerosing BCCs are classified as distinct histological variants in the WHO classification of skin tumors [20]. In the present study, we did not classify sclerosing BCCs as a distinct histological variant as the clinical and microscopic features of both variants in human medicine are highly similar. Additionally, the limited number of infiltrating BCCs did not allow for the identification of distinguishing features. Identification and characterization of additional cases from affected reptiles would be necessary to allow their future classification as distinct histological variants.

Superficial BCCs are a highly prevalent histological variant in humans that presents as a well-defined, red, scaly patch or plaque with a central clearing and thin rolled edges [21]. This histological variant of BCCs was not identified in this study and has not been previously reported in other animal species. It should be considered, however, that the diagnosis of superficial BCCs may be easily missed in reptiles and other animals due to its discrete gross pathological features and low tendency to invade or ulcerate, in comparison to other BCC histological variants that mostly have a prominent invasive nodular appearance. Consequently, these lesions are presumably often not routinely submitted for histological evaluation or misdiagnosed as SCCs in situ or non-neoplastic lesions [43].

In the present study, a certain discriminative value of IHC staining using E-cadherin and COX-2 towards differentiating SCCs from BCCs, including KA from well-differentiated SCCs and keratotic BCCs from other BCC histological variants, was demonstrated. The use of epithelial antigen clone Ber-EP4 and EMA IHC staining does not seem to have a value towards distinguishing reptilian SCCs and BCCs as immunoreactivity was absent, in contrast to their highly discriminative value in human SCCs and BCCs. The latter can presumably be attributed to lack of cross-reactivity with mammal antibodies or different protein expression patterns between reptilian and mammal tissues [24]. With regard to EMA, it has been documented that the EMA gene emerged during evolution from poikilothermic reptiles to homoeothermic mammals and for this reason, the epitopes for the EMA antibodies are presumably lacking in reptiles [44]. It might have been interesting to explore the value of CD10 and Bcl-2 markers in reptiles, as they are highly reliable markers to distinguish SCCs from BCCs in humans [27,45]. However, a similar lack of immunoreactivity, as observed for epithelial antigen clone Ber-EP4 and EMA markers, is to be expected [46,47,48].

In humans and dogs, it has been demonstrated that upregulated COX-2 in SCCs enhances prostaglandin synthesis, which increases cell proliferation, promotes angiogenesis, inhibits immunosurveillance, and enhances invasiveness [38,49,50]. Based on the results of this study, it can be presumed that COX-2 overexpression in reptile BCCs and SCCs, as previously demonstrated in KAs in lizards, can also be used as a marker for invasiveness and it might be interesting to explore the development of therapeutic strategies relying on the effect of specific COX-2 enzyme inhibitors for the adjuvant treatment of SCCs and keratotic BCCs in reptiles [8,15]. Intercellular adhesion is mediated by E-cadherin, which is typically downregulated in SCCs with high invasive and metastatic potential [51,52]. Despite evidence in humans and dogs indicating that BCCs are generally less invasive than SCCs [12,51,52], BCC histological variants with low E-cadherin expression in squamates and chelonians, such as solid, keratotic, and infiltrative BCCs, may exhibit unusually rapid and infiltrative growth, as reported in a terminal BCC in a Hermann’s tortoise [16]. Further studies, correlating the IHC staining patterns with the clinical course and therapeutic results are needed to further elucidate the value of these IHC markers.

## 5. Conclusions

Based on the established human, canine, and feline classifications for skin neoplasms, a classification for SCCs and BCCs, and their histological variants, in squamates and chelonians is proposed in the present study. This proposed classification should be subjected to continuous evaluation and revision as new scientific insights emerge. Accurate histological classification of SCC and BCC histological variants is considered crucial to predict their biological behavior and guide treatment decisions, particularly for histological variants with high invasiveness and metastatic potential. The results of the present study allow the correct identification of SCCs and BCCs in squamates and chelonians and might be highly valuable in diagnosing and differentiating challenging histological variants, such as keratotic BCCs and BCSCCs. IHC staining with E-cadherin and COX-2 markers aids in the differentiation of SCCs from BCCs and their histological variants in squamates and chelonians. COX-2 overexpression in reptile SCCs implies that specific COX-2 enzyme inhibitors could be included as a part of the adjuvant therapy of these neoplastic disorders. Further research should warrant the association of the distinct histological variants of SCCs and BCCs of squamates and chelonians to their respective biological behavior.

## Figures and Tables

**Figure 1 animals-13-01327-f001:**
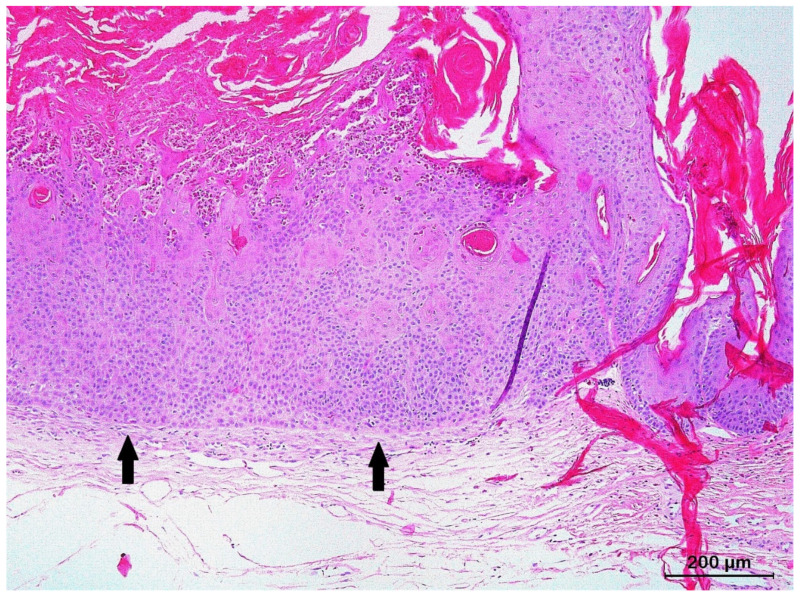
Histological section of a squamous cell carcinoma in situ from the skin of the eyelid of a panther chameleon (*Furcifer pardalis*). Prominent epidermal dysplasia with enlarged and pleomorphic squamous cells replacing the entire thickness of the epidermis can be noted. The basal membrane remains intact (arrows), without invasion of the dermis.

**Figure 2 animals-13-01327-f002:**
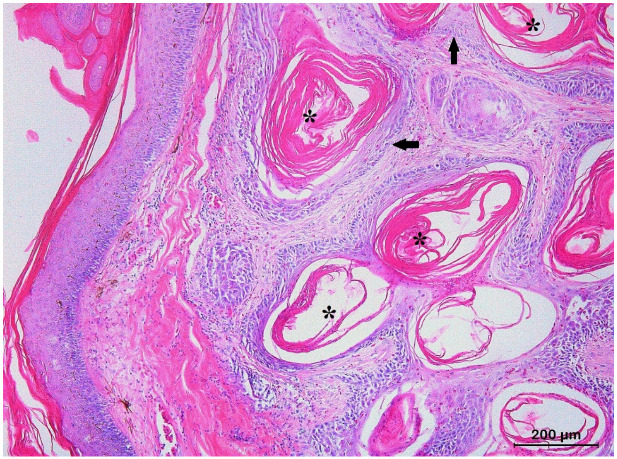
Histological section of a well-differentiated squamous cell carcinomas (SCCs) from the skin of the body wall in a bearded dragon (*Pogona vitticeps*) with tumor cells containing abundant eosinophilic cytoplasm. Synchronous differentiation of peripheral basal-type cells to central squamous epithelial cells (arrows) can be observed, eventually resulting in the formation of keratin pearls (asterisks).

**Figure 3 animals-13-01327-f003:**
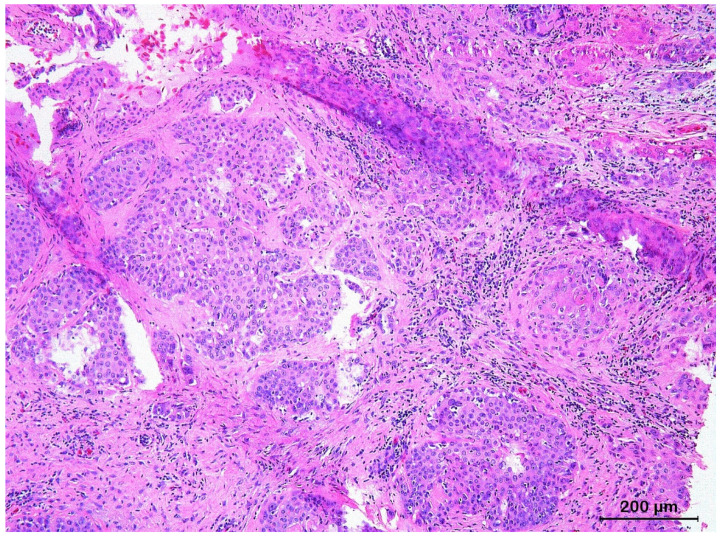
Histological section of a moderately differentiated squamous cell carcinoma from the oral mucosa of an African spurred tortoise (*Centrochelys sulcata*) showing tumor cells containing scant eosinophilic cytoplasm, haphazard squamous differentiation and marked nuclear atypia.

**Figure 4 animals-13-01327-f004:**
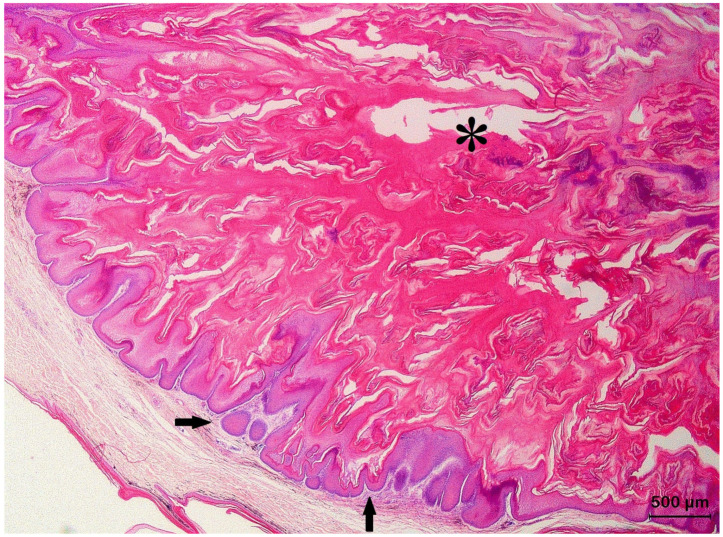
Histological section of a keratoacanthoma from the skin of the body wall of a bearded dragon (*Pogona vitticeps*) showing a characteristic architectural pattern consisting of an exo–endophytic, cyst-like invagination of the epidermis that creates a crateriform lesion with a central keratinous plug (asterisk) and minimally infiltrating borders (arrows).

**Figure 5 animals-13-01327-f005:**
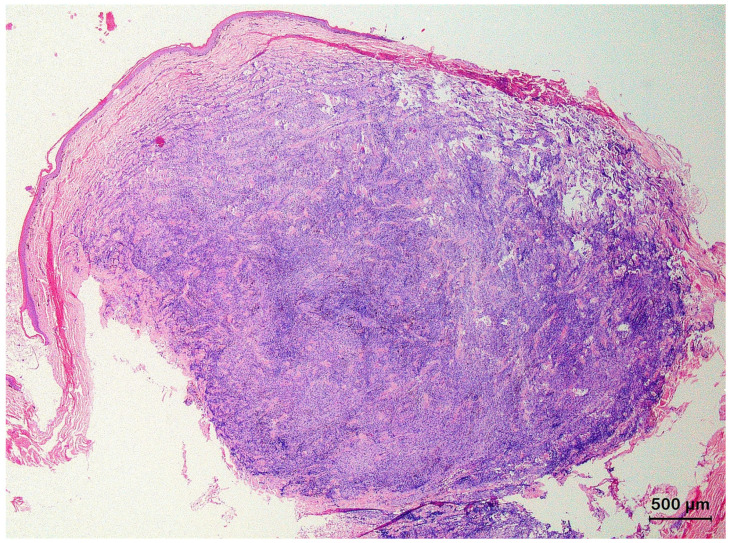
Histological section of a solid basal cell carcinoma originating from the skin of the front leg of a red-eared slider (*Trachemys scripta elegans*) characterized by a nodular pattern composed of cords of small polyhedral basaloid cells extending into the dermis.

**Figure 6 animals-13-01327-f006:**
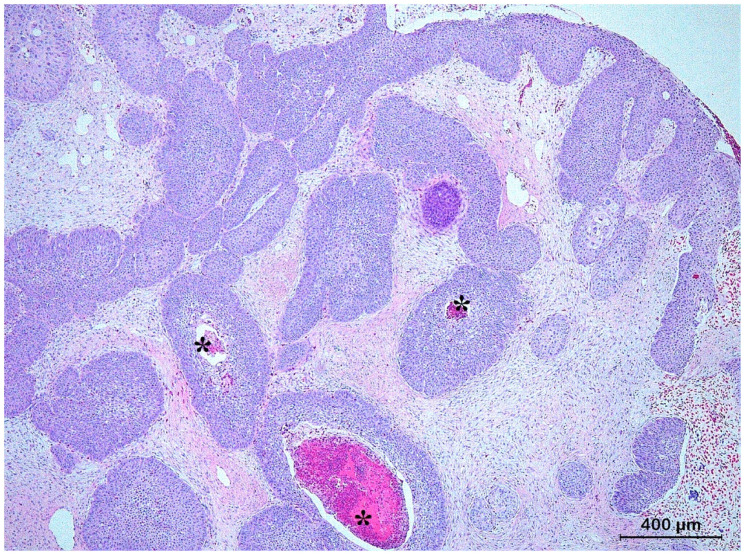
Histological section of a keratotic basal cell carcinoma from the skin of the body wall of a boa constrictor (*Boa constrictor*) characterized by cords of small polyhedral basaloid cells extending into the dermis and foci of abrupt squamous differentiation with central mature keratinization (asterisks).

**Figure 7 animals-13-01327-f007:**
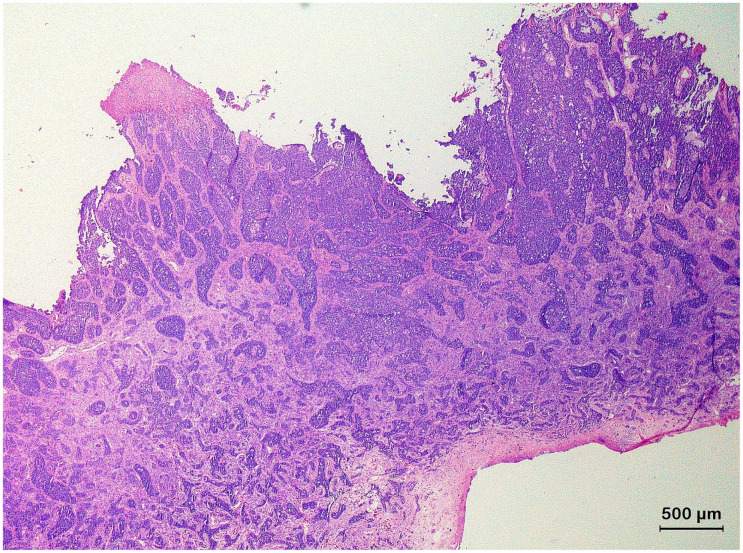
Histological section of an infiltrating basal cell carcinoma originating from the epidermis of the shell of a Hermann’s tortoise (*Testudo hermanni*) characterized by irregular, narrow, and elongated cords of small, atypical basophilic basal tumor cells.

**Figure 8 animals-13-01327-f008:**
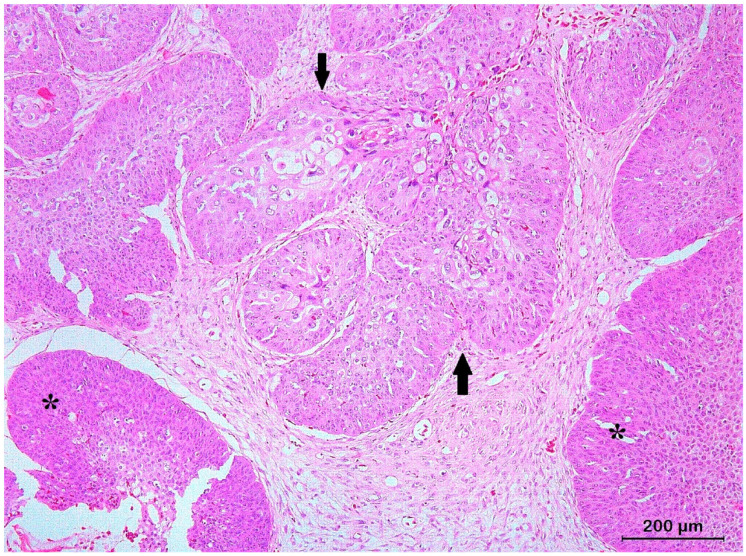
Histological section of a basosquamous cell carcinoma originating from the epidermis of the shell of a Hermann’s tortoise (*Testudo hermanni*) presenting as an invasive front showing BCCs (asterisks) as well as SCCs (arrows) histological features.

**Figure 9 animals-13-01327-f009:**
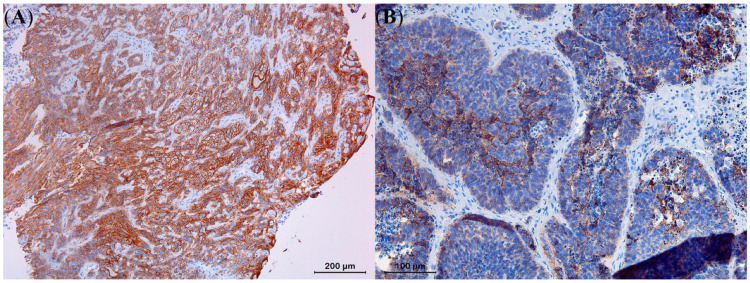
Strong E-cadherin expression with immunoreactivity at the level of the plasma membrane could be noted in all moderately differentiated squamous cell carcinoma cells (**A**), in contrast to moderate to poor expression levels in cells of a solid basal cell carcinoma (**B**).

**Figure 10 animals-13-01327-f010:**
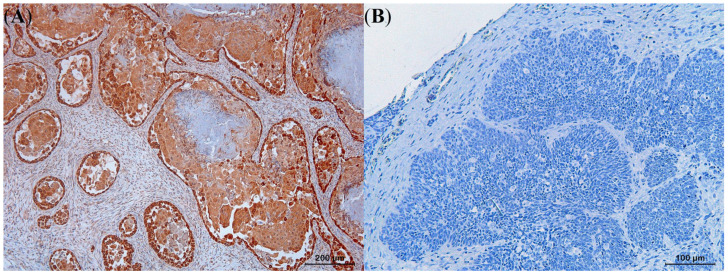
Diffuse and strongly positive intracytoplasmic cyclooxygenase-2 expression could be noted in a well-differentiated squamous cell carcinoma (**A**) in contrast to negative cyclooxygenase-2 expression in a solid basal cell carcinoma (**B**).

**Table 1 animals-13-01327-t001:** Anatomic location and number of tumors from 35 reptile patients that were initially diagnosed as squamous cell carcinomas (SCCs) or basal cell carcinomas (BCCs) and re-classified into histological variants following retrospective histological and immunohistochemical characterization.

Species	Location	Initial Diagnosis	Re-Classification	
			Final Diagnosis	Histological Variant
Bearded dragon (*Pogona vitticeps*) (*n* = 9)	Skin	6 SCC	4 SCC	3 WD SCC
				1 KA
			1 BCC	1 Infiltrating BCC
			1 Non-neoplastic	Gingival fibrous hyperplasia
	MCJ	2 SCC	2 SCC	1 WD SCC
				1 MD SCC
	Oral	1 BCC	1 BCC	1 Solid BCC
Panther chameleon (*Furcifer pardalis*) (*n* = 5)	Skin	4 SCC	4 SCC	4 KA
	MCJ	1 SCC	1 SCC	1 SCC in situ
Veiled chameleon (*Chamaeleo calyptratus*) (*n* = 2)	Skin	1 SCC	1 SCC	1 KA
	MCJ	1 SCC	1 BCC	1 Keratotic BCC
Brown anole (*Anolis sagrei*) (*n* = 2)	Skin	2 SCC	1 SCC	1 WD SCC
			1 BCC	1 Keratotic BCC
Common blue-tongued skink (*Tiliqua scincoides*) (*n* = 1)	Skin	1 SCC	1 Non-neoplastic	Cystic mass lined by pseudostratified ciliated epithelium
Green iguana (*Iguana iguana*) (*n* = 1)	Skin	1 SCC	1 BCC	1 Keratotic BCC
Von Höhnel’s chameleon (*Trioceros hoehnelii*) (*n* = 1)	Skin	1 SCC	1 BCC	1 Keratotic BCC
Boa constrictor (*Boa constrictor*) (*n* = 1)	Skin	1 SCC	1 BCC	1 Keratotic BCC
False map turtle (*Graptemys pseudogeographica*) (*n* = 1)	Skin	1 SCC	1 Non-neoplastic	Pyogranulomatous dermatitis
Yellow-belied slider (*Trachemys scripta scripta*) (*n* = 1)	Skin	1 SCC	1 SCC	1 WD SCC
Hermann’s tortoise (*Testudo hermanni*) (*n* = 9)	Skin	1 SCC	1 BCC	1 Infiltrating BCC
	Shell	2 SCC	1 SCC	1 WD SCC
		3 BCC	4 BCC	1 Solid BCC
				2 Infiltrating BCC
				1 BSCC
	MCJ	1 SCC	1 SCC	1 MD SCC
	Oral	2 BCC	2 BCC	2 Solid BCC
African spurred tortoise (*Centrochelys sulcata*) (*n* = 1)	Oral	1 SCC	1 SCC	1 MD SCC
Red-eared slider (*T. scripta elegans*) (*n* = 1)	Skin	1 BCC	1 BCC	1 Solid BCC

MCJ, mucocutaneous junction of the eyelid; WD, well-differentiated SCC; MD, moderately differentiated SCC; KA, keratoacanthoma; BSCC, basosquamous cell carcinoma.

**Table 2 animals-13-01327-t002:** Immunoreactive score system (IRS) [36].

A (Percentage of Positive Cells)	B (Intensity of Staining)	IRS Score (A × B)
0 = no positive cells	0 = no color reaction	0–1 = negative expression
1 ≤ 10% positive cells	1 = mild reaction	2–3 = poor expression
2 = 10–50% positive cells	2 = moderate reaction	4–8 = moderate expression
3 = 51–80% positive cells	3 = intense reaction	9–12 = strong expression
4 ≥ 80% positive cells	Final IRS score (A × B): 0–12 ^1^	

1 The immunoreactive score (IRS) is calculated by multiplying the positive cells proportion score (0–4) and the staining intensity score (0–3).

**Table 3 animals-13-01327-t003:** Histological characteristics of 17 squamous cell carcinomas (SCCs) and 15 basal cell carcinomas (BCCs) histological variants obtained from squamates and chelonians. The mitotic index and degree of nuclear atypia are specified for each histological variant.

Neoplasm	Total	Mitotic Index			Degree of Nuclear Atypia		
		0–2	3–4	≥5	<30%	30–60%	>60%
SCC SCC in situ Conventional SCC ◦ WD ◦ MD KA	17	15 (88.2%)	1 (5.9%)	1 (5.9%)	11 (64.7%)	4 (23.5%)	2 (11.8%)
	1	0	0	1	1	0	0


	7	6	1	0	3	4	0
	3	3	0	0	1	0	2
	6	6	0	0	6	0	0
BCC Solid Keratotic Infiltrating BSCC	15	7 (46.7%)	7 (46.7%)	1 (6.7%)	8 (53.3%)	3 (20%)	4 (26.7%)
	5	4	1	0	4	0	1
	5	2	2	1	4	1	0
	4	1	3	0	0	2	2
	1	0	1	0	0	0	1

WD, well-differentiated SCC; MD, moderately differentiated SCC; KA, keratoacanthoma; BSCC, basosquamous cell carcinoma.

**Table 4 animals-13-01327-t004:** Expression of E-Cadherin and COX-2 according to the IRS score system (0–12) in 17 squamous cell carcinomas (SCCs) and 15 basal cell carcinomas (BCCs) from squamate and chelonian species.

Neoplasm	Total	E-Cadherin (IRS Score) (%)					COX-2 (IRS Score) (%)			
		Neg	Poor	Mod	Strong	Neg	Poor	Mod	Strong
SCC	17	0 (0%)	0 (0%)	5 (29.4%)	12 (70.6%)	0 (0%)	0 (0%)	10 (58.8%)	7 (41.2%)
SCC in situ	1	0	0	0	1	0	0	0	1
Conventional SCC									
◦ WD	7	0	0	5	2	0	0	3	4
◦ MD	3	0	0	0	3	0	0	1	2
KA	6	0	0	0	6	0	0	6	0
BCC	15	0 (0%)	3 (20%)	12 (80%)	0 (0%)	7 (46.7%)	4 (26.7%)	4 (26.7%)	0 (0%)
Solid	5	0	1	4	0	4	1	0	0
Keratotic	5	0	1	4	0	0	1	4	0
Infiltrating	4	0	1	3	0	3	1	0	0
BSCC	1	0	0	1	0	0	1	0	0

WD, well-differentiated SCC; MD, moderately differentiated SCC; KA, keratoacanthoma; BSCC, basosquamous cell carcinoma; Neg, negative; Mod, moderate.

## Data Availability

The data presented in this study are available on request from the corresponding author.

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
