# Peer review of "Histological Variants of Squamous and Basal Cell Carcinoma in Squamates and Chelonians: A Comprehensive Classification"

_animals, 2023, doi:10.3390/ani13081327_

Round 1

Reviewer 1 Report

Dear authors, I have only one question and recommendation for your manuscript. In lines 77 - 80 you mentioned that The discrimination of dermal SCC and BCC and their histological variants is highly important towards prognosis estimation and establishing appropriate treatment protocols as the associated invasiveness, recurrence rate, and metastatic potential are strongly correlated with the involved histological variant [15,16].

I think you should improve this part of manuscript with short description of different treatment protocols/approaches to different variants of BCC and SCC (in reptiles?), like the authors you cited, Paolino et al, Burton et al, did. I think this would increase the interest to read you paper for clinicians (non academic, not scientists). This is my only comment to your  manuscript. 

Author Response

Dear authors, I have only one question and recommendation for your manuscript. In lines 77 - 80, you mentioned that “The discrimination of dermal SCC and BCC and their histological variants is highly important towards prognosis estimation and establishing appropriate treatment protocols as the associated invasiveness, recurrence rate, and metastatic potential are strongly correlated with the involved histological variant[15,16]”.

- I think you should improve this part of the manuscript with a short description of different treatment protocols/approaches to different variants of BCC and SCC (in reptiles?), like the authors you cited, Paolino et al, Burton et al, did.

We appreciate the remark of the reviewer. The main objective of this study was to provide a comprehensive histological and immunohistochemical characterization of SCC and BCC in reptiles, including their different histological variants. Although we acknowledge the importance of understanding different treatment protocols and approaches for various histological variants of BCC and SCC, we deemed it beyond the scope of this study. In a forthcoming research article that will be the second part of the present article, we will delve into the biological behaviour of each histological variant in reptiles, as well as the optimal treatment protocols and prognosis for each case, being able to clarify all doubts and questions that may arise from the clinicians. It is worth noting that our decision to not discuss different treatment protocols for the various histological variants in this paper was due to the lack of data in previously published reviews, research articles, and case reports in reptiles. These sources did not identify the histological variants being treated, making it challenging to add short descriptions about the different treatment protocols without adding our own data and starting an entire discussion about it.

I think this would increase the interest in reading your paper for clinicians (non-academic, not scientists). This is my only comment to your manuscript.

Reviewer 2 Report

It has been a pleasure to read this manuscript. It is really interesting as the authors characterize and classify differect types of tumors in squamates and chelonians. The Introduction place the study in context very clearly. Material and Methods are well described and give enough details. The Results are also precise and give data enough to carry out the histological classification proposed in this paper. I think that such a classification is interesting to have a better understanding of different clinical situations. There is only one point that should be considered. The authors state that the number of patients was 35. But then, in table 1, it is not clear the number of individuals of each species. In my opinion it should be corrected.

Author Response

It has been a pleasure to read this manuscript. It is really interesting as the authors characterize and classify different types of tumours in squamates and chelonians. The Introduction places the study in context very clearly. Material and Methods are well described and give enough details. The Results are also precise and give data enough to carry out the histological classification proposed in this paper. I think that such a classification is interesting to have a better understanding of different clinical situations. There is only one point that should be considered.

The authors state that the number of patients was 35. But then, in Table 1, it is not clear the number of individuals of each species. In my opinion, it should be corrected.

We appreciate the remark of the reviewer. Table 1 has been changed as suggested by the reviewer.